# Combinatorial Delivery of Gallium (III) Nitrate and Curcumin Complex-Loaded Hollow Mesoporous Silica Nanoparticles for Breast Cancer Treatment

**DOI:** 10.3390/nano12091472

**Published:** 2022-04-26

**Authors:** Thimma Mohan Viswanathan, Vaithilingam Krishnakumar, Dharmaraj Senthilkumar, Kaniraja Chitradevi, Ramakrishnan Vijayabhaskar, Velu Rajesh Kannan, Nachimuthu Senthil Kumar, Krishnan Sundar, Selvaraj Kunjiappan, Ewa Babkiewicz, Piotr Maszczyk, Thandavarayan Kathiresan

**Affiliations:** 1Department of Biotechnology, Kalasalingam Academy of Research and Education, Krishnankoil 626126, India; viswasonofmohan9@gmail.com (T.M.V.); senthil.gene4@gmail.com (D.S.); chitradevik11@gmail.com (K.C.); sundarkr@klu.ac.in (K.S.); selvapharmabio@gmail.com (S.K.); 2Department of Microbiology, Bharathidasan University, Tiruchirappalli 620024, India; krishnavasan@gmail.com (V.K.); uvrajesh@gmail.com (V.R.K.); 3Department of Surgical Oncology, Meenakshi Mission Hospital and Research Centre, Madurai 625107, India; drvijayabhaskar@gmail.com; 4Department of Biotechnology, Mizoram University, Aizawl 796004, India; nskmzu@gmail.com; 5Department of Hydrobiology, Faculty of Biology, University of Warsaw, 02-089 Warsaw, Poland; ewa.babkiewicz@wp.pl (E.B.); fizbanek@wp.pl (P.M.)

**Keywords:** anticancer, cell viability, drug loading capacity, drug release, mitochondrial protein, nanomedicine

## Abstract

The main aims in the development of a novel drug delivery vehicle is to efficiently carry therapeutic drugs in the body’s circulatory system and successfully deliver them to the targeted site as needed to safely achieve the desired therapeutic effect. In the present study, a passive targeted functionalised nanocarrier was fabricated or wrapped the hollow mesoporous silica nanoparticles with 3-aminopropyl triethoxysilane (APTES) to prepare APTES-coated hollow mesoporous silica nanoparticles (HMSNAP). A nitrogen sorption analysis confirmed that the shape of hysteresis loops is altered, and subsequently the pore volume and pore diameters of GaC-HMSNAP was reduced by around 56 and 37%, respectively, when compared with HMSNAP. The physico-chemical characterisation studies of fabricated HMSNAP, Ga-HMSNAP and GaC-HMSNAP have confirmed their stability. The drug release capacity of the fabricated Ga-HMSNAP and GaC-HMSNAP for delivery of gallium and curcumin was evaluated in the phosphate buffered saline (pH 3.0, 6.0 and 7.4). In an in silico molecular docking study of the gallium-curcumin complex in PDI, calnexin, HSP60, PDK, caspase 9, Akt1 and PTEN were found to be strong binding. In vitro antitumor activity of both Ga-HMSNAP and GaC-HMSNAP treated MCF-7 cells was investigated in a dose and time-dependent manner. The IC_50_ values of GaC-HMSNAP (25 µM) were significantly reduced when compared with free gallium concentration (40 µM). The mechanism of gallium-mediated apoptosis was analyzed through western blotting and GaC-HMSNAP has increased caspases 9, 6, cleaved caspase 6, PARP, and GSK 3β(S9) in MCF-7 cells. Similarly, GaC-HMSNAP is reduced mitochondrial proteins such as prohibitin1, HSP60, and SOD1. The phosphorylation of oncogenic proteins such as Akt (S473), c-Raf (S249) PDK1 (S241) and induced cell death in MCF-7 cells. Furthermore, the findings revealed that Ga-HMSNAP and GaC-HMSNAP provide a controlled release of loaded gallium, curcumin and their complex. Altogether, our results depicted that GaC-HMNSAP induced cell death through the mitochondrial intrinsic cell death pathway, which could lead to novel therapeutic strategies for breast adenocarcinoma therapy.

## 1. Introduction

Hollow mesoporous silica nanoparticles (HMSNP) are highly versatile and favorable nanoplatforms for drug delivery systems (DDS) because of their assortment, accessibility, and biocompatibility. HMSNP is non-lethal and has enormous explicit surface territory, a tunable pore size, excellent physicochemical steadiness, and artificially modifiable surfaces [1,2]. Enhancing the targeted drug delivery of HMSNPs to specific cells/tumor tissues might be adopted in two main strategies. One is the selected appropriate gatekeepers, which are attached to pore openings of drug loaded HMSNPs by covalent linking [3]. The various compounds that are reported in silica nanoparticles are coated with polymers, such as polyethylene glycol (PEG), polycaprolactone (PCL), dextran, chitosan, polyethyleneimine (PEI), and 3-aminopropyl triethoxysilane (APTES), to enhance its drug loading, retention time, and release and prevention of aggregation. The other strategies are the drug molecules linked to either the pore or surface of the HMSNPs through stimulus responsive linkage [4,5]. The HMSNP is an efficient nanocarrier for enabling greater storage and release of chemical substances and chemotherapeutic drugs [6,7,8]. Silicon is naturally found in bones and connective tissues of the human body [9]. The cytotoxicity and cellular uptake of the MSNs depend on the size and the surface charges of the nanocarrier. The 15 nm diameter nanocarriers induced more cytotoxicity than 100 nm nanocarriers in endothelial cells [10]. The mesoporous silica nanoparticles are loading large amounts of multiple drugs, which are double responsive to the synergistic therapy of breast cancers [11]. Subsequently, chemotherapeutic drugs, such as doxorubicin, are combined with proapoptotic peptide [12], curcumin [13], and Bcl-2 siRNA [14].

Until now, several chemotherapeutic drugs have been used against breast cancer through the alteration of histone deacetylase inhibitors and various cell surface receptors, like EGFR and HER2, to inhibit cancer survival pathways. An efficient cancer treatment strategy is the combination of multiple drugs which synergistically interact against cancer cells but not in normal cells [15]. In this regard, curcumin and its subsidiaries express anticancer activity by stifling the multiplication of various tumour cell lines [16,17]. Specifically, curcumin deregulates GRP78, IRE1α, and CHOP, inducing cell death [13]. Gallium (Ga) is the second metal compound after platinum [18] that is considered a high potential candidate for anticancer therapy [19,20,21]. Gallium compounds bind with transferrin and are taken up by an endocytic process of cells through the transferrin receptor to form transferrin-gallium (Tf-Ga) complexes [22,23]. The Tf-Ga disturbs the normal uptaking of the transferrin-iron complex, which leads to iron deprivation in cancer cells [24]. Moreover, intracellular gallium compounds distort the three-dimensional structure of DNA and its replication and protein synthesis, and inhibit ATPase, DNA polymerase and tyrosine-specific protein phosphatase activity to induce cell death [25]. The gallium-based metallodrugs, such as gallium nitrate, gallium chloride, gallium maltolate, tris (8-quinolonato) gallium (III) or KP46, have been used as anticancer agents. Among them, gallium maltolate and KP46 have been evaluated in preclinical trials [26,27], and gallium III nitrate has gone up to clinical trial phase II [28]. Gallium compounds are also used in combination with other chemotherapeutic agents, such as paclitaxel [21], gemcitabine [29], vinorelbine [30], hydroxyurea [24], fludarabine [31], and interferon-γ [32], and are found to synergistically inhibit various types of cancers. They are directed to assess wide ranges of antineoplastic movement and inhibition of tumor growths of Hodgkin’s and non-Hodgkin’s lymphoma, chronic lymphocytic leukemia, prostate, lung, ovarian, cervical, bladder, renal, breast, melanoma, and sarcoma [20].

The combination of gallium (III) nitrate and curcumin therapy has not yet been attempted, and this is the first study to analyse their interaction as well as the mechanism behind gallium-curcumin complex-induced cancer cell death. The present study focuses on cetyltrimethyl ammonium bromide (CTAB) as a template and tetraethyl ortho-orthosilicate (TEOS) as a silicon source for synthesised HMSNPs. The (3-aminopropyl) triethoxysilane (APTES) were hydrolysed, and NH_2_ was crippled on the surface of HMSNP to form HMSNAP, which facilitates binding through the amination reaction in the gallium and hydroxyl group of enolic in curcumin. HMSNAP is used as a carrier for delivering gallium (III) nitrate alone (Ga-HMSNAP) and in combination with gallium (III) nitrate-curcumin (GaC-HMSNAP) complex against MCF-7 cells. The key questions asked in the present study are: (1) whether HMSNAP is less toxic and has rapid drug loading capacity and sustained drug release in MCF-7 cells; (2) whether Ga-HMSNAP and GaC-HMSNAP can induce cell death in MCF-7 cells at a low gallium concentration; and (3) which proteins are up-regulated or down-regulated due to GaC-HMSNAP treatment in MCF-7 cells. Overall, our studies portrayed an improvement in the therapeutic potential of the combination of gallium and curcumin (GaC-HMSNAP), demonstrating that they could be a better alternative for the current therapies against cancer.

## 2. Materials and Methods

### 2.1. Materials and Reagents

The chemicals Gallium (III) nitrate (99.90% purity), curcumin (99.50% purity), tetraethyl orthosilicate (TEOS) (99.00% purity), 3-Aminopropyl triethoxysilane (APTES) (99.00% purity), cetyltrimethyl ammonium bromide (CTAB) (99.00% purity), dimethyl sulfoxide (DMSO) (99.70% purity), acridine orange (98.90% purity), ethidium bromide (99.50% purity), and casein from bovine milk (99.00%) was procured from Sigma-Aldrich, (St. Louis, MO, USA). Dulbecco’s modified eagle medium (DMEM), Foetal bovine serum (FBS), and 0.25% trypsin EDTA (Gibco BRL, Waltham, MA, USA) and Tetrazolium salt (3-(4,5-dimethylthiazol-2yl)-2,5-diphenyltetrazolium bromide) (98.00% purity) was obtained from (Invitrogen, Carlsbad, CA, USA). BCA reagent was received from Bio-Rad (Hercules, CA, USA). All the experimental solutions were prepared using 18 MΩ milli-Q water 133 (Millipore system, Burlington, MA, USA). A few other analytical grade chemicals and reagents were purchased from Thermo Fisher Scientific Ltd. Mumbai, India.

### 2.2. Synthesis of HMSNAP

The sol-gel method was adopted for the synthesis of hollow mesoporous silica nanoparticles (HMSNP) with slight modifications [33]. The formation of HMSNP was attained as ammonia effectuate silica precursor underwent rapid hydrolysis with polycondensation of an ethanol-water mixture by using surfactant CTAB as a structure-directing agent to assist in the formation of the mesostructure during the sol-gel process. A 40.5 mL mixture of an ethanol-water containing 80 mg of CTAB and 500 μL of TEOS was added and mixed well, followed by 500 μL of liquid ammonia, and stirred for 3 h at 700 g. After 3 h, the resultant white precipitate products were washed with ethanol and dried. The dried HMSNP were calcinated in the air at 800 °C for 6 h. Functionalisation with amines, 50 mg HMSNP were added in 50 mL ethanol and 50 µL APTES and kept in a shaker for 24 h, then washed three times with ethanol and one time with distilled water to form HMSNAP.

### 2.3. Drug Loading and Release

For drug loading, 15 mg of synthesised HMSNAP were dispersed separately with 30 to 50 mM of gallium alone, and a combination of 30 to 50 mM gallium with 30 mM curcumin was dissolved in ethanol in an orbital shaker for 24 h. The unbound drug was washed with ethanol and dried to obtain the nanoparticles, and various concentrations of the drug-loaded nanocarrier were used in the cell viability assay to identify the IC_50_ value of drugs and their combination. For the drug loading and release assay, 15 mg of synthesised HMSNP and HMSNAP were dispersed separately with 36 mM Gallium (III) nitrate alone and in the combination of 36 mM Gallium (III) nitrate with 30 mM curcumin added in each 10 mL of ethanol in an orbital shaker for 48 h. The unbound free gallium and curcumin were estimated for each 6 h interval from 0 h to 48 h incubation. Their absorbance at 540 nm using a fluorescent microplate reader (Biotek, Winooski, VT, USA) was compared with control. The amount of gallium and curcumin-loaded nanostructure was determined by the formula (Drug loading = O.D value at 0 h − O.D value at different time intervals/O.D value at 0 h) × 100.

The drug release experiments were accomplished with 2 mg mL^−1^ of 30 μM Ga-HMSNAP. A combination of 20 μM Gallium and 30 μM curcumin-loaded GaC-HMSNAP was kept separately in a phosphate buffer saline (PBS) in three different pH: 3, 6 and 7.4. Approximately 200 μL of the sample was collected at every 6 h interval up to 72 h, and the absorption of gallium release at 420 nm and the curcumin release at 540 nm was determined using a microplate reader.

### 2.4. Drug Release Kinetics

The gallium from HMSNAP and the gallium-curcumin complex from HMSNAP release data were separately fitted into five different kinetics models (zero, first, Higuchi, Korsmeyer-Peppas, and Hixson-Crowell) to describe the mechanism of the drug release kinetics and diffusion. The release kinetic mechanisms were evaluated by using the above investigated in vitro release data through the DD solver 1.0 tool (Microsoft-Excel plugin model). The zero-order rate kinetics explains the rate of drug release and does not relate to their concentrations. The first-order rate kinetics explains the drug release from the formulation where the drug release rate related to their concentration is dependent. The Higuchi model designated the drug release from the insoluble matrix as a square root of the time-dependent process based on Fickian diffusion. The Korsmeyer-Peppas model explains a simple mathematical relationship, describing the drug release from a formulation [34,35].

### 2.5. Characterisation of HMSNAP

The synthesised silica nanoparticles were powdered and analysed by SEM-EDAX, TEM, dynamic light scattering (DLS), zeta potential, X-ray diffraction (XRD), and Fourier transform infrared spectroscopy (FTIR). To find out the morphological structure of HMSNAP, Ga-HMSNAP and GaC-HMSNAP were loaded on carbon paper for SEM (Evo18 Zeiss, Munich, Germany) and copper grid for TEM (HRTEM, T12 tecnai, Hillsboro, OR, USA) at HT650 ES1000W t 120 kV. The pore size of this HMSNAP was measured by using the Image J software (National Institutes of Health, New York, NY, USA) of HRTEM. The nanostructures are dispersed in water and analysed for their size and surface charge through DLS and zeta potential using the Nanoparticle analyser SZ-100 (Horiba, Kyoto, Japan). XRD analysis was done in an X-ray tube 3 kW with a copper target, real-time multiple strip solid-state detectors were used, K alpha was maintained at 0.001°, and it was scanned using the D8 Advance ECO XRD System (Bruker, Madison, WI, USA). FTIR measurements of the above nanoparticles were analysed using a Shimadzu spectrometer (Nishinokyo, Japan) in the ranges of 4000–400 cm^−1^ in the transmission mode.

The surface elemental composition of the nanoparticles was analysed with X-ray photoelectron spectroscopy (VersaProbe III Scanning XPS Microprobe, Physical electronics, Chanhassen, MN, USA), applying an emission line of aluminium Kα x-ray source with hv of 1486.6 (eV), carried out with an emission current of 10 mA and an anode voltage of 15 kV. The samples mounted on the gold layers sputtered onto the silicon substrate. Data were analysed by using MultiPak Data Reduction Software (version 9.6.0) (2500 Hagisono, Chigasaki, Japan). Nitrogen adsorption-desorption isotherm analysis was carried out by Quantachrome^®^ ASiQwin™ (Boynton Beach, FL, USA) at liquid nitrogen (−196 °C). Before measurement, the samples were degassed at 150 °C under vacuum conditions for 24 h. The specific surface area (SBET) of nanocarriers was calculated from the relative pressure range of nitrogen adsorption from 0.05 to 0.2 using the Brunauer–Emmett–Teller (BET) equation. Pore size distributions (PSD) were determined from adsorption twigs of isotherms using the Barrett–Joyner–Halenda (BJH) method. The total pore volume (Vp) was estimated from the amount of nitrogen adsorbed at a relative pressure of 0.99. The drugs that bound with the nanocarriers were evaluated by ^1^ H nuclear magnetic resonance (NMR). The spectra were measured on a Bruker AV 500 MHz spectrometer with D_2_O as the solvents and tetramethylsilane (TMS) as the internal standard. The chemical structure of nanoparticles with drug loading was determined through the use of TopSpin^®^ 4.0 software.

### 2.6. Molecular Docking Analysis

Molecular docking was performed using AutoDock 4.2 software (The Scripps Research Institute, San Diego, CA, USA). The basic and default settings were followed for protein and ligand preparation. Docking of the gallium-curcumin complex ligand with proteins, such as PDI, calnexin, HSP60, PDK caspase 9, Akt1, and PTEN, was performed. The binding model for each protein was analysed using a Discovery Studio Visualizer v4.0 [36].

### 2.7. Cell Culture and Cell Viability Assay

The breast adenocarcinoma cell line MCF-7 was procured from NCCS (National Centre For Cell Science, Pune, Maharashtra, India). The MCF-7 cells was grown in Dulbecco’s modified eagle medium (DMEM), which consisted of 10% fetal bovine serum (FBS) and 5% CO_2_ at 37 °C. The MCF-7 cell viability was determined through MTT assay, as described previously, with slight modifications [37]. The cells were trypsinised from T25 confluence flasks and seeded in 96 well plates at 15,000 cells per well. The different concentrations (5 to 50 µg) of HMSNAP, Ga-HMSNAP, and GaC-HMSNAP were treated separately with MCF-7 cells and incubated at different time intervals for 24, 48 and 72 h with a serum (0.5%) deprivation medium. The cell viability of nanoparticles on MCF-7 cells was analysed using tetrazolium salt. After respective time incubation, 10 µL of MTT solution was added to each well and incubated for 4 h. The tetrazolium reacted with dehydrogenase reductase of live cells to produce indissoluble formazan crystal. Then 100 µL DMSO was added to each well to dissolve the formazan crystal to form purple formazan, and the concentration was read at 595 nm in a microtiter plate reader. The experiments were repeated three times with results and were expressed as the percentage (%) of control cells. The percentage of cell viability and IC_50_ values of each drug were calculated using Prism-GraphPad version 5.

### 2.8. Measurement of Apoptosis by Acridine Orange/Ethidium Bromide (AO/EtBr)

The apoptosis assay was carried out with AO/EtBr double staining, in HMSNAP, Ga-HMSNAP and GaC-HMSNAP-treated MCF-7 cells. The drug-loaded nanoparticle-treated cells were trypsinised and pelleted after 24, 48, and 72 h incubation periods. The MCF-7 cells were dissolved in 1 mL of phosphate buffer saline containing 100 µL each of acridine orange (50 µg mL^−1^) and ethidium bromide (30 µg mL^−1^), and were examined under a fluorescent microscope (Carl Zeiss, Oberkochen, Germany). Randomly selected various fields were analysed and the percentages of live and dead cells were estimated.

### 2.9. Western Blot Analysis

The qualitative analyses of free gallium, Ga-HMSNAP, and GaC-HMSNAP induced protein alteration in MCF-7 cells were analysed using western blotting. The cells were grown in 100 mm dishes and treated with 5µg HMSNAP, and an IC_50_ concentration of 40 µM free gallium, 35µM Ga-HMSNAP, and combination of 25 µM gallium and 30 µM curcumin-loaded GaC-HSMNAP for 24 and 48 h incubation. After the respective incubation times, MCF-7 cells were harvested and lysed with 50 mM phosphate buffer (pH 7.4) containing 25 mM sodium fluoride, 5 mM EDTA, 0.4% ASB-14, and protease and phosphatase cocktail inhibitors (Roche, Switzerland) on ice for 5 min. and the lysates were sonicated and followed by centrifugation at 12,000 rpm for 10 min. at 4 °C. Next, the protein was quantified with BCA reagent. Each 50 µg of proteins were loaded onto SDS-PAGE and proteins were run at 90 V for 2 h. The proteins were transferred to nitrocellulose membrane (Amersham Bioscience, Piscataway, NJ, USA). Anti-cleaved PARP, anti-caspase 12, anti-cleaved caspase 12, and 6, anti-caspases 9, 6, anti-calnexin, anti-PDI, anti-prohibitin, anti-HSP 60, anti-SOD1, anti-phospho-Akt (S473), anti-total Akt, anti-phospho-PTEN (S380), anti-phospho-c-Raf (S259), anti-phospho-PDK1 (S241), anti-phospho-GSK3β (S9), and anti-β-actin were obtained from Cell Signaling Technology (Danvers, MA, USA). The membrane was blocked with 5% casein for 1 h and incubated with primary antibodies for 1 h at room temperature followed by either anti-rabbit IgG or anti-mouse IgG HRP-linked secondary antibodies (Santa Cruz, CA, USA) for 1 h. The presence of immunoreactive bands of respective proteins was detected with the addition of lumiglo (Thermo scientific, Rockford, IL, USA). All of the immunoreactive and phosphorylation signals were quantified using the densitometric scanner (Bio-Rad, Hercules, CA, USA). The blot was washed and reprobed with anti-β-actin to ensure that an equal amount of proteins were loaded in each well and the data presented were representative of the two independent experiments with similar results.

### 2.10. Statistical Analysis

Each experiment was repeated three times with three replicates, and the standard error of the mean (SEM) values were optimised for all experimental data as ±mean. A one-way analysis of variance test was performed to assess the results between the control and all treated samples (*p* < 0.05).

## 3. Results and Discussion

### 3.1. Synthesis and Characterisation of HMSNAP

The schematic representation of synthesis and drug loading of HMSNAP is as shown in Figure 1a. HMSNAP was synthesised and loaded with different concentrations of either gallium alone or with a combination of gallium and curcumin. The drug-loaded HMSNAP was analysed using SEM, TEM, EDAX, DLS, and zeta potential in Figure 1. The SEM and TEM analysis illustrated that pores are opened in APTES decorated HMSNAP. Both the drug-loaded Ga-HMSNAP and GaC-HMSNAP pores were closed and concentrated in the hollow and mesopore (Figure 1b–e) of the nanocarrier; this could be due to the drastic loading of gallium and curcumin in HMNSAP. The particle size analysis (Figure 1h) of HMSNAP, Ga-HMSNAP, and GaC-HMSNAP exhibited sizes 180, 186 and 188 nm, respectively. Therefore, no significant changes of particle size were observed in HMSNAP and drug-loaded HMSNAP. The EDAX analysis suggested that HMNSAP contains Si-16.34%, and O-55.1%, along with GaC-HMNSAP holding C-46.43%, Si-27.22%, O-24.72%, and Ga-1.66% (Figure 1f,g). The presence of O indicates surface functionalisation with APTES followed by carbon and gallium, which indicates that curcumin and gallium are loaded in HMSNAP.

The zeta potential of the nanocarrier was further analysed and net surface charges are exhibited in Figure 1i. The surface charges of HMNSAP, Ga-HMSNAP, and GaC-HMSNAP were +25.4, −20.3 and −18.1 mV, respectively. The surface charges are reduced due to the gallium and gallium-curcumin complex loaded on the nanocarrier surface. The negative charge of the zeta potential demonstrated that Ga-HMSNAP and GaC-HMSNAP are highly stable and that their aggregation is inhibited. Cancer research has established that the surface-charge assessments of cancer cells in terms of zeta potential have been found to be a useful biological feature in identifying cellular interactions with certain nanomaterials. Therefore, the zeta potential ranges between +30 to −30 mV has a high degree of stability of the nanoparticles and they help in the sustained drug release inside the cells [2].

The physical properties of synthesised HMSNAP, Ga-HMSNAP, and GaC- HMSNAP were investigated and compared. The X-ray diffraction (XRD) curves of the HMSNAP, Ga-HMSNAP, and GaC-HMSNAP are presented in Figure 2a. These results suggest that the XRD pattern of HMSNAP showed a broader peak at 2θ conditions 12°, which indicates the amorphous nature. The XRD spectra of Ga-HMSNAP exhibited broader peaks at 2θ conditions 12°, 20° and 30°, while the XRD spectra of GaC-HMSNAP showed a sharper peak at 2θ conditions 10°, 20° and 30°, these results indicated the amorphous nature for the two drug-loaded HMSNAP. FTIR investigation revealed that peaks around 500 and 1800 cm^−1^ confirmed the nearness functional group of Si–O–Si, C–H (weak), N–H, and C=O in the silica nanocarrier in Figure 2b. In HMSNAP the most intensive band of Si–O–Si bending is present in the range of 500–800 cm^−1^. The peak value indicates the symmetric and anti-symmetric stretching vibration of the Si–O–Si. The C=N and OH gathering and peaks were found in the scopes of 2200 to 3780 cm^−1^, which showed that functional group C=N is from the gallium nitrate and OH is from the curcumin loaded in HMSNAP. Subsequently, huge disparities were found in Ga-HMSNAP and GaC-HMSNAP when contrasted with HMSNAP. The peak range of 2943 cm^−1^ showed the nearness of the amine group and –C–H group peak range of 2830 cm^−1^, which indicates the functional group APTES.

The surface element analysis of HMSNAP and drug-loaded HMSNAP were assessed by XPS. The survey XPS spectra of HMSNAP, Ga-HMSNAP, C-HMSNAP, and GaC-HMSNAP were observed from 0 to 1100 eV, as shown in Figure 3a. The elements of Si, O, C and N were detected in the spectrum of HMSNAP. The elements like Si and O are evolved from the hydrolysis of TEOS, whereas the C and N represent the APTES surface coating of the nanocarrier. Compared with the spectrum of HMSNAP, the new element Ga appeared in the sample of Ga-HMSNAP, suggesting that the gallium immobilised on the surface of HMSNAP through aminated APTES. In addition, a huge amount of C appeared in the sample of C-HMSNAP and GaC-HMSNAP, providing persuasive evidence for the surface modification of HMSNAP. On the other hand, the intensity of Si and O was further decreased after surface modification with gallium and curcumin. All of the above XPS results indicated that we successfully modified HMSNAP alone and in a combination of gallium and curcumin.

The high-resolution XPS spectra of Si2p, N1s, O1s, Ga3d, and C1s were shown in Figure 3b–f. More importantly, the peak intensities of Si2p in HMSNAP, Ga-HMSNAP, C-HMSNAP, and GaC-HMSNAP are shown in Figure 3b and the binding energy of Si2p was located in between 101 to 103.8 eV. The intensity of drug-loaded HMSNAP of Si2p peak is around 42% reduced when compared with HMSNAP. The N1s spectra were displayed in Figure 3c and the ranges from 396.4 to 402.6. The intensity of N1s in Ga-HMSNAP, C-HMSNAP, and GaC-HMSNAP was decreased when compared with HMSNAP. In particular, the N1s spectrum in the sample HMSNAP shows a sharp peak and the others demonstrate blunt peaks. The intensity of O1 ranged from 531 to 533.6 eV, and high intensity was found in HMSNAP and gradually decreased in C-HMSNAP, GaC-HMSNAP and Ga-HMSNAP in Figure 3d. Therefore, the O1 is masked by drugs loading on the nanocarrier. The high-resolution spectra of Ga3d are from 18.8 to 23.6 and two different peaks appeared in the spectra. The intensity of spectra is reduced in GaC-HMSNAP when compared with a gallium-alone-loaded nanocarrier in Figure 3e. In the spectra of C1s, the peak position of C1s in HMSNAP, Ga-HMSNAP, C-HMSNAP, and GaC-HMSNAP is shown in Figure 3f and the binding energy of C1s ranged from 270.8 to 273.8 eV. The percentages of elements in HMSNAP, Ga-HMSNAP, C-HMSNAP, and GaC-HMSNAP were calculated via the XPS spectra, as shown in Table 1. The percentages of Si, N, O, and C in HMSNAP are 22.4, 0.4, 55.8, and 21.4%, respectively. It is worth mentioning that the percentages of Si and O were close to the expected ratio of Si/O (1:2). Compared with HMSNAP, the percentages of Si, N, O, Ga, and C gain Ga-HMSNAP are changed to 12.8, 6.0, 31.6, 4.4, and 45.2%, respectively. The increase of N content suggests that APTES immobilised on the surface of HMSNAP through hydrolysis reaction. Moreover, the element Ga with 4.4% emerged in the sample of Ga-HMSNAP. It is demonstrated that gallium is loaded in the nanocarrier. Similarly, C-HMSNAP containing Si (13%), C (44.2%), O (32.2%), and N (6.6%) and GaC-HMSNAP consist of Si (12.4%), C (46.4%), N (1.8%), O (39.4%), and Ga (4%). When compared to HMSNAP, C is abundantly expressed in both C-HMSNAP and GaC-HMSNAP, which is represented by curcumin and loaded in the nanocarrier.

In addition, N_2_ adsorption-desorption isotherms of HMSNAP and drug-loaded HMSNAP is shown in Figure 4, and they have demonstrated a typical H1 hysteresis loop, which proves their mesoporous structure [31] The isotherm of HMSNAP and drug-loaded HMSNAP displayed monolayer absorption at the relative pressure (P/P^0^) ranges from 0.2–1.0 [32]. The relative pressure (P/P^0^) of HMSNAP, Ga-HMSNAP, C-HMSNAP and GaC-HMSNAP are 0.8, 0.3, 0.8, and 0.2, respectively, as shown in Figure 4a–d. The drug-loaded nanocarrier shape of the hysteresis loops is changed when compared with the control, which indicated that the absorption and pore volume of the nanocarrier was altered due to the drug loading of the nanocarrier. The Brunauer–Emmett–Teller (BET) surface area and pore volume of HMSNAP are 1067.78 m^2^ g^−1^, and 3.25 cm^3^ g^−1^, respectively, as shown in Table 2. After introducing gallium and curcumin separately into the HMSNAP, the BET surface area is 710.07 cm^3^ g^−1^ and 946.89 cm^3^ g^−1^, and the pore volume is 1.49 cm^3^ g^−1^ and 2.48 cm^3^ g^−1^, respectively, and both are reduced when compared with control. Furthermore, the gallium-curcumin complex loaded HMSNAP surface area is 637.39 cm^3^ g^−1^ and the pore volume is cm^3^ g^−1^. Both the drug-loaded nanocarriers surface and pore volume are further reduced because of the amination of gallium and carboxylation of curcumin with the nanocarrier, making it suitable for a large amount of drug-loading efficiency and smart delivery vehicles. Likewise, the Barrett–Joyner–Halenda (BJH) pore diameters are gradually reduced in Ga-HMSNAP (3.11) and C-HMSNAP (3.35), and further reduced GaC-HMSNAP (2.80) when compared with control (4.45), as shown in the insets in Figure 4a–d. Furthermore, the specific surface area, SBET around 37% of pore diameters and 56% of pore volumes are filled with drug loading. What is noteworthy to mention here is that the pore diameters of the control and drug-loaded nanocarriers could be interpreted as such that the drugs are successfully embedded homogeneously in the skeleton of mesoporous silica structure and the unique features of SiO_2_ have not been destroyed.

The comprehensive information of the drug-loaded complex structure of HMSNAP is investigated using the ^1^H-NMR spectrum, as shown in Figure 5a. The functional group of silica –O–Si (2.65) of HMSNP and N-H (5.11) of APTES are successfully immobilised to the surface of silica through the hydrolysis between APTES and the hydroxyl group on the silica sphere surface to form HMNSAP (Figure 5a). Consequently, –O–Si (1.75 and 2.65) confirm that the gallium binds with the nanocarrier and the chemical shifts overshade the amine functionalisation of the surface sphere of silica. In addition, N–H from APTES (5.11) is masked by gallium loading on the nanocarrier in Figure 5b. Besides that, the O–C of curcumin (3.83 and 3.90) are bound with –O–Si (3.55), the surface of silica through methyl and methine group formation and the enolic group formation in curcumin. Further aromatic groups of (C–OH 5.35, 6.99, and 7.12) confirm the presence of the nano-curcumin complex in Figure 5c. In the formation of the gallium-curcumin complex, the functional groups of gallium are altered and bound through C=O (1.54 and 1.79) of the keto-enol group of curcumin. The O–C (3.83 and 3.90) of curcumin is bound with –O–Si (3.55) of silica through the hydroxyl group of enolic in Figure 5d. The silica nanocarriers bind with the curcumin of gallium-curcumin complex, but not in the gallium bound on the silica surface, because curcumin has a more functional group when compared with gallium. All of the above information concerning the gallium-curcumin complex indicates that the intensity of curcumin is slightly higher than the gallium-nano complex in HMSNAP. These experiments confirmed that the drugs are loaded in both the hollow and mesopore of the nanocarrier through alteration of surface area, pore size, pore volume and surface charges, the addition of compounds, and its functional groups. Therefore, the explicit volumes of drugs are loaded into the nanocarrier for the therapy of breast cancer.

### 3.2. Drug Loading and Release in HMSNP and HMSNAP

The gallium alone and combination of gallium and curcumin are separately dissolved in 100% ethanol then dispersed with HMSNP and APTES-coated HMSNAP separately and evaluated for their drug-loading capacity, as shown in Figure 6a. The amount of drugs loaded in amine-functionalised HMSNAP was determined from an unbound drug in a flask and was measured at every 6 h interval and up to 48 h of incubation. Drugs loaded on amine-functionalised HMSNAP were found to be 88%, however 24% of drug loading are observed in HMSN. Therefore, APTES is an efficient encapsulation with silica nanoparticles and enhances its drug loading capacity. Nonetheless, HMSNAP was functionalised with APTES, which hampered the penetration of gallium and curcumin from the inner part of HMSNAP. The loading of gallium and curcumin in the nanocarrier presumably comes mainly from the electrostatic interactions of their outer surface that significantly enhances the increase in loading ability [38].

The loaded drug materials in HMSNP and HMSNAP were evaluated for their release properties at normal pH 7.4 and acidic pH 3.0 and 6.0 with phosphate buffered saline in Figure 6b. The drugs were gradually released from the nanocarrier, depending on the increased incubation time. The gallium and curcumin complex were 87.5% released in pH 7.4, 69.57% in pH 6.0 and 59.24% in pH 3 at 72 h. Similarly, 81.3, 71.15, and 60.25% of gallium released in pH 7.4, 6.0, and 3.0 at 72 h, respectively. Therefore, gallium and curcumin were released effectively at normal pH when compared with the acidic pH range. These results suggested that a higher amount of drugs are retained in the reservoir of HMSNAP and released gradually with pH and in a time-dependent manner. The amount of drug released depends on the degradability of APTES coating with HMSNAP. Therefore, the ethoxy group of APTES is hydrolysed so that a higher volume of drugs can be released from the carrier [39].

### 3.3. Drug Release Kinetics

In the present study, for both formulations, the drug release kinetics data is depicted in Table 3. The release rate constant and regression coefficient (r^2^) value were predicted from each drug release kinetic model. In general, with a regression coefficient value (r^2^) near 1, the relationship between the two factors are great to fit. The regression coefficient value (r^2^) for release pattern of zero-order kinetics are much closer to 1 for both the kinetics models, i.e., for gallium 0.9565 (pH 3), 0.9561 (pH 6), 0.9546 (pH 7.4) and the gallium-curcumin complex 0.9653 (pH 3), 0.9697 (pH 6), 0.9763 (pH 7.4) (source file attached in Appendix A), respectively. Simultaneously, in the Korsmeyer-Peppas model, the r^2^ value of gallium alone and the gallium-curcumin complex was higher than 0.98. From the regression coefficient value (r^2^), the results indicate that the controlled release of gallium and gallium-curcumin from the nanocarrier and the release kinetics mechanism were adopted through Fickian diffusion. Based on these release kinetic models, the zero-order kinetics model is the best model for the gallium and gallium-curcumin release mechanism. These results also indicate the uniform dissolution and controlled release of the nanoparticle formulation.

### 3.4. Molecular Docking

To determine whether the gallium-curcumin complex ligand directly interacts with western blot, resulting in proteins, we performed a molecular docking analysis with seven proteins in Figure 7. The seven proteins, including PDI, calnexin, HSP60, PDK caspase 9, Akt1, and PTEN, are directly associated with the gallium-curcumin complex and the remaining proteins are indirectly associated and regulated with the complex ligand. The docking results revealed a strong binding energy between the complex ligand and protein molecules, such as PDI (−5.42 kcal mol^−1^), calnexin (−7.04 kcal mol^−1^), HSP60 (−6.34 kcal mol^−1^), PDK (−8.55 kcal mol^−1^), caspase 9 (−6.63 kcal mol^−1^), Akt1 (−9.84 kcal mol^−1^), and PTEN (−3.89 kcal mol^−1^). PDI, PDK, Akt, calnexin, and caspases 9 have two hydrogen bond interactions and HSP60, and PTEN has one hydrogen bond between the ligand complex and other associated proteins. A molecular analysis revealed the presence of several non-covalent interactions and strong hydrogen bonds. These interactions between the associated proteins and the gallium-curcumin ligand complex could contribute to the regulation of breast cancer.

### 3.5. Cell Viability Effect of HMSNAP and Drug-Loaded HMSNAP

MCF-7 cells were treated with various concentrations of HMSNAP, free gallium, Ga-HMSNAP, and GaC-HMSNAP and their IC_50_ value was assessed via MTT assay as shown in Figure 8a. The 5 µg and 10 µg HMSNAP induced 2 to 3% and 4 to 6% cell death when incubated at 24 to 72 h, respectively. APTES coating on HMSNAP has significantly reduced its toxicity in MCF-7 cells. Therefore, all the drugs were loaded in the 2 to 4 µg of HMSNAP used in all experiments. The IC_50_ value of curcumin in MCF-7 cells was 30 µM, which was previously described by our group [13]. Ga-HMSNAP and GaC-HMSNAP-induced MCF-7 cell death was directly proportional to dose-and time-dependent manner. The IC_50_ value at 24 h was 45 µM for free gallium, 40 µM for Ga-HMSNAP, and 30 for µM GaC-HMNSAP in MCF-7 cells, as shown in Figure 8b. At 48 h incubation, 40 µM free gallium, 35 µM Ga-HMSNAP and 25 µM GaC-HMSNAP induced 50% cell death, as shown in Figure 8c. At 72 h, 35 µM gallium, 30 µM Ga-HMSNAP, and 20 µM GaC-HMNSAP induced 50% of cell death, as shown in Figure 8d. The significance in traditional cancer medicine with HMSNAPs is highly conventional and non-toxic. In breast cancer, the HMNSAPs used to overcome the formal issues can applied in effective attention of significant in HMSNAPs [45]. Therefore, the IC_50_ value of the gallium-curcumin complex is reduced by 40% gallium when compared with the free gallium concentration. Consequently, the combination of curcumin and nanocarrier with gallium has effectively induced cell death at a lesser gallium concentration.

### 3.6. Evaluation of Apoptosis in HMSNAP and Drug-Loaded HMSNAP on MCF-7 Cells

Apoptosis plays a major role when the cells cannot sustain their homeostasis during cell maintenance and its proliferation. Dysregulation of apoptosis is known to trigger in the development of cancer [44]. Therefore, the induction of cancer cell death is a major focus in formulating anticancer drugs. In general, GaC-HMSNAP is an effective therapeutic molecule and induces the apoptosis of cancer cells. Herein, GaC-HMNSAP-mediated cell death altered morphological changes and this was analysed by acridine orange/ethidium bromide dual staining, as shown in Figure 9a. The control and HMNSAP-treated MCF-7 cells exhibited green fluorescence, and the nuclei were round and homogenous, indicating viable cells. IC_50_ concentrations of free gallium, Ga-HMNSAP, and GaC-HMNSAP-treated cells at 24 h incubation presented as orange, which indicates the early-stage of apoptosis. In addition, red fluorescence cells containing apoptotic bodies, indicating necrotic cells and the complete loss of membrane integrity, were found in Ga-HMNSAP and GaC-HMNSAP-treated cells at both 48 h and 72 h. The percentage of early, late apoptosis and necrosis of drug-treated MCF-7 cells are shown in Figure 9b. HMSNAP-treated cells showed 3 to 7% of cell death occurring in 24 h to 72 h of incubation. The free gallium, Ga-HMSNAP, and GaC-HMSNAP-treated cells were found 27 to 41%, 31 to 50%, and 40 to 56%, respectively, from 24 h to 72 h. Significant necrosis was found in GaC-HMSNAP at 48 h and free gallium (17.22%), Ga-HMSNAP (21.93), and GaC-HMSNAP (23.52%) at 72 h. The Ga-HMSNAP and GaC-HMSNAP-treated cancerous cells have induced elevation of ROS production, which leads to oxidative stress in cells [18]. This increased ROS production is repealed by the mitochondrial antioxidant protein mitoquinone, leading to cell death [46]. In addition, gallium alters membrane permeability and mitochondrial functions [47].

### 3.7. Ga-HMSNAP and GaC-HMSNAP Induce Apoptosis through ER and Mitochondrial Target Proteins

Furthermore, to understand the mechanism of gallium-mediated apoptosis, we initially analysed the proteins involved in ER homeostasis in MCF-7 cells, as shown in Figure 10a,b. Free gallium, Ga-HMSNAP, and GaC-HMSNAP-treated MCF-7 cells showed a 1.5-fold increased expression of calnexin at 24 h, whereas it drastically reduced two- to three-fold at 48 h. The overexpressed calnexin helps maintain cell proliferation, whereas suppression of calnexin is known to induce cell death through prolonged ER stress and caspases-mediated cleavage [48,49]. The protein disulfide isomerase (PDI) gradually decreased two-to three-fold at 48 h when compared with the control. PDI is highly abundant and upregulated in many cancer types, and overexpression of PDI may serve as a diagnostic marker for metastasis. PDI inhibitors have a therapeutic role against cancer progression [50]. Our results also suggested that GaC-HMSNAP inhibition of PDI and calnexin could lead to an increase in ER stress and activation of cleaved caspases (Figure 10e,f), resulting in induced cell death in human breast cancer [51,52].

The mitochondrial proteins prohibitin 1 and HSP 60 are three- to four-fold downregulated at 48 h in Ga-HMSNAP and GaC-HMSNAP treated MCF-7 cells when compared with the control, as shown in Figure 10a,b. Prohibitin 1 is located in the mitochondria and plasma membrane and is predominantly involved in maintaining mitochondrial morphology and its functions. Similarly, the knockdown of prohibitin 1-induced the cell death in the ovarian [53] and hepatocarcinoma cell lines [54]. The HSP 60 is highly expressed in human breast cancer when compared with normal tissues; this enhances cancer cell proliferation, and HSP 60 inhibitors downregulate HSP 60, thereby inducing the apoptosis of tumour cells [55]. Therefore, it can be concluded that gallium and curcumin complex downregulates prohibitin 1 and HSP 60, and then enhances MCF-7 cell death through an intrinsic apoptotic pathway. SOD1 is aberrantly expressed in cancerous tissues and promotes cancer cell proliferation [56]. The Ga-HMSNAP and GaC-HMSNAP-treated MCF-7 cells reduced SOD1 activity and induced cell death approximately three-fold. Therefore, gallium-mediated SOD1 inhibition could suppress the proliferation of cancer cells through the downregulation of PTEN and Akt (Figure 10c,d) [57] and the upregulation of caspases 3 and 9 (Figure 10e,f) [58]. SOD1 inhibition induces apoptosis of cancer cells through activation of the mitochondria-dependent apoptotic pathway.

### 3.8. Ga-HMSNAP and GaC-HMSNAP Inhibit Phosphorylation of Tumourigenic Proteins and Induce Phosphorylation of Tumour-Suppressor Proteins

Further mechanistic investigation of the gallium-curcumin complex-induced cell death was done by analysing the alteration of tumourigenic and tumour-suppressor proteins in MCF-7 cells, as shown in Figure 10c,d. The free gallium and Ga-HMSNAP and GaC-HMSNAP-treated MCF-7 cells showed phosphorylation of Akt at S473, and the total Akt is downregulated two-fold at 48 h incubation when compared with the respective control. Moreover, our results confirmed that gallium-curcumin complex-treated cells downregulate prohibitin 1 (Figure 10a,b), leading to the reduced phosphorylation of Akt and the inhibition of cell proliferation [59]. The free gallium, Ga-HMSNAP, and GaC-HMSNAP-treated cells demonstrated 2.5-fold reduction of phosphorylation of PTEN at S380 compared to the untreated cells (Figure 10c,d). The suppression of PTEN in MCF-7 cells inhibited cell proliferation and induced apoptosis by reducing Akt and induced the activation of ATP and caspases [60]. The immunoblot study reveals that c-Raf phosphorylation at S249 is fully inactivated by free gallium-treated MCF-7 cells at both 24 and 48 h. However, Ga-HMSNAP and GaC-HMSNAP-treated MCF-7 cells gradually reduced up to four-fold in 48 h-treated cells when compared with the control. C-Raf is localised in the mitochondrial membrane and plasma membrane and is phosphorylated at S259, inactivated by Akt and negatively regulates the Raf activity, inducing cell death [61]. Prohibitin 1, along with Akt, is specifically involved in the Ras-Raf-MEK-ERK pathway. PDK1 promotes cancer through the activation of Akt and the downstream of PDK-1 may be a promising therapeutic target of breast cancer [62]. Therefore, our results clearly suggest that downregulation of PHB1 leads to inactivation of phosphorylation of S471 of Akt, S259 of c-Raf, and S241 of PDK1 and induces the intrinsic cell death pathway in MCF-7. In the western blot analysis, Ga-HMSNAP and GaC-HMSNA-treated MCF-7 cells showed the activation of phosphorylation of GSK3β at S9. GSK3β at S9 was gradually increased up to three- to four-fold in gallium, Ga-HMSNAP and GaC-HMSNAP-treated MCF-7 cells when compared with control. GSK 3β is involved in cell cycle regulation through phosphorylation of cyclin D1. The phosphorylation of S9 deactivates the GSK 3β. Therefore, our results showed that phosphorylation of GSK 3β at S9 attenuates Akt and PKC, and then induces cell death [63].

### 3.9. GaC-HMSNAP Induces Mitochondrial and Other Apoptotic Proteins

PARP is a nuclear protein that is mainly involved in the repair of damaged DNA. It is cleaved by caspases to form cleaved PARP and is unable to repair DNA, thus triggering apoptosis of the cells. Western blot analysis of the c-PARP, cleaved caspase 12, and caspase 6 showed two- to 2.5-fold increases in Ga-HMSNAP and GaC-HMNSAP-treated MCF-7 cells when compared with the relevant control (Figure 10e,f. The PARP is cleaved by caspases and is considered to be a hallmark of apoptosis [64,65,66]. Therefore, Ga-HMSNAP and GaC-HMSNAP induced cell death by promoting c-PARP. However, AO/ETBr stained gallium and curcumin complex-treated MCF-7 cells showed ruptured plasma membranes due to the increase of c-PARP. Caspase 12, cleaved caspase 12, caspase 6 and cleaved caspase 6 expressions increased 1.5-fold upon Ga-HMSNAP and GaC-HMNSAP treatment as compared to the control. No significant variations are observed in HMSNAP-treated cells with untreated cells in all apoptotic proteins. The lesser concentration of HMSNAP is non-toxic. Therefore, our results clearly illustrate that HMSNAP does not induce any cell-death-regulated proteins.

## 4. Conclusions

In summary, the present study investigates the co-delivery efficacy of gallium (III) nitrate alone and in combination with gallium and curcumin-loaded hollow mesoporous silica nanoparticles in MCF-7 cells. The drug-loaded nanomaterials are characterised through FTIR, XPS, nitrogen sorption, ^1^ H-NMR, etc., confirming that both of the drugs are immobilised on the surface of HMSNAP during aminated APTES. In addition, an enormous amount of C was found in the drug-loaded nanocarrier, providing convincing proof for the surface modification of HMSNAP. On the other hand, the intensity of Si and O was further decreased after surface modification with gallium and curcumin. Around 37% of pore diameters and 56% of pore volumes are occupied by the drug-loaded nanocarrier. All of the above XPS, nitrogen sorption, and ^1^H-NMR results indicated that we have successfully modified HMSNAP alone and in combination with gallium and curcumin. The mechanism of drug delivery is triggered by the signalling proteins, which induce cell death in MCF-7 cells. HMSNAP was non-toxic and no significant cell death occurred in MCF-7. The IC_50_ values of the combination of gallium and curcumin (20 µM) are significantly reduced proto-oncoproteins that were downregulated in GaC-HMNSAP-treated cells. Therefore, it is reasonable to conclude that gallium and curcumin interact with these phosphoproteins and induce cell death. The ER client proteins calnexin and PDI are drastically reduced in GaC-HMNSAP-treated MCF-7 cells, indicating that GaC-HMNSAP disturbs ER homeostasis. The western blot analysis clearly demonstrated that GaC-HMSNAP interacts with oncoproteins and mitochondrial proteins, inducing an intrinsic cell death pathway. Further docking studies of gallium and curcumin complexes indicate the interaction of these molecules with the apoptotic and mitochondrial proteins through hydrogen bonding formation and with a higher binding energy that could lead to cell death. Altogether, our results clearly depicted that GaC-HMNSAP altered tumour suppressor and mitochondrial proteins and induced the intrinsic cell death pathway.

## Figures and Tables

**Figure 1 nanomaterials-12-01472-f001:**
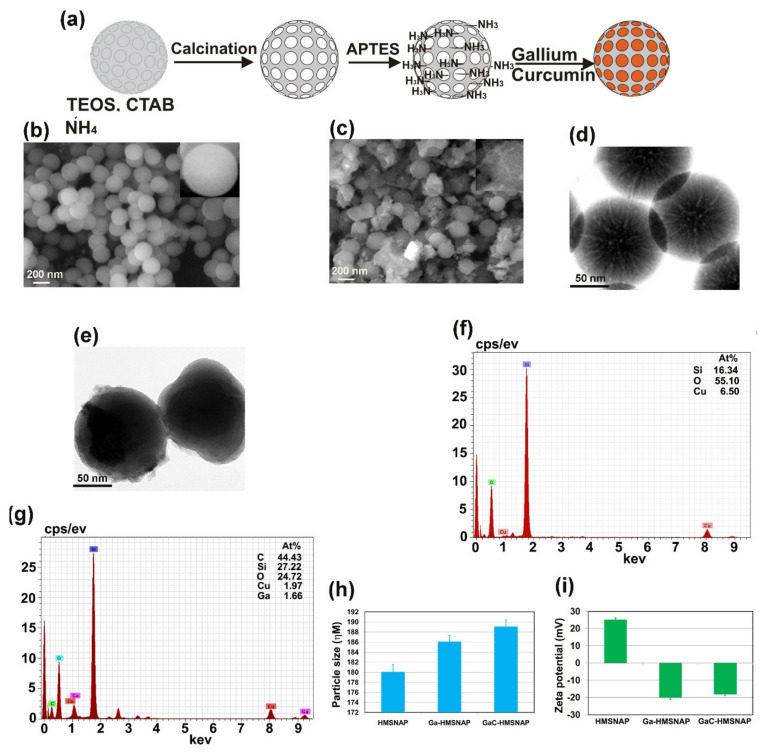
(**a**) Schematic illustration of synthetic procedure for HMSNAP and encapsulation of gallium and curcumin-loaded nanocarrier (**a**). SEM and TEM images of HMSNAP and drug-loaded HMSNAP (**b**–**e**), EDAX, the pattern of HMSNAP and drug-loaded HMSNAP (**f**,**g**), particle size and zeta potential measurement of HMSNAP, Ga-HMSNAP, and GaC-HMSNAP (**h**,**i**).

**Figure 2 nanomaterials-12-01472-f002:**
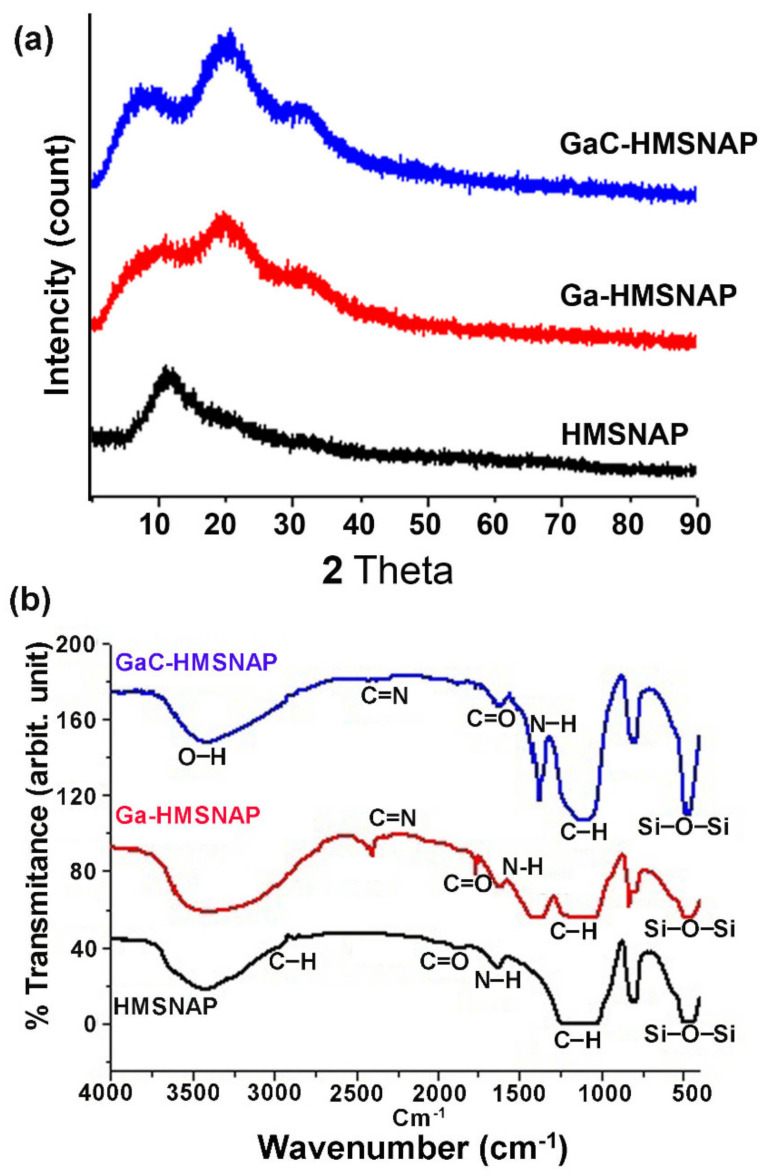
(**a**) XRD and (**b**) FTIR pattern of HMSNAP and drug−loaded HMSNAP.

**Figure 3 nanomaterials-12-01472-f003:**
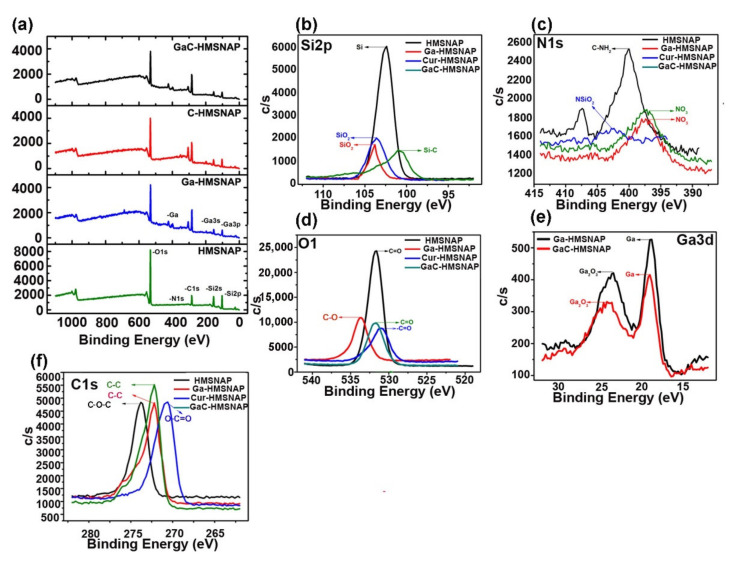
The XPS spectra of survey (**a**) and high resolution of HMSNAP and drug-loaded HMSNAP (**b**–**f**). The region of Si2p (**b**), the region of N1s (**c**), the region of O1s (**d**), the region of Ga3d (**e**), the region of C1s (**f**).

**Figure 4 nanomaterials-12-01472-f004:**
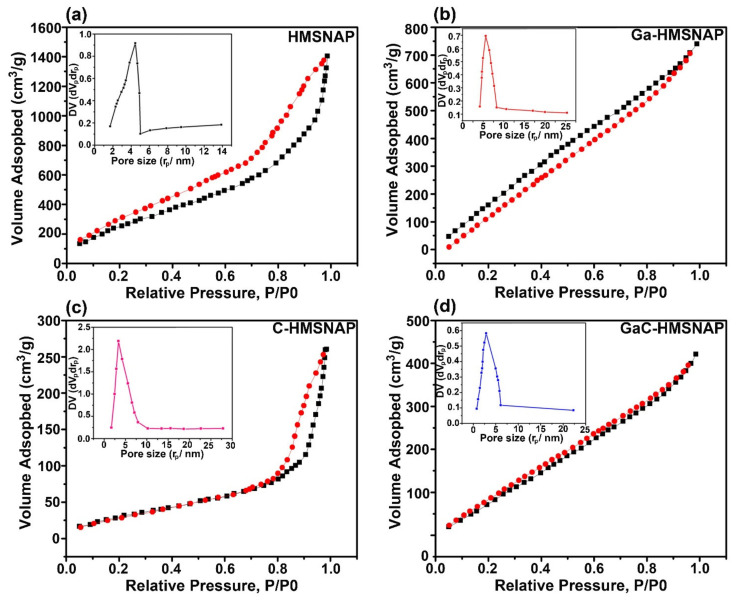
Nitrogen adsorption-desorption isotherms and pore-size distribution (insets figure) of HMSNAP (**a**), Ga-HMSNAP (**b**), C-HMSNAP (**c**), and GaC-HMSNAP (**d**).

**Figure 5 nanomaterials-12-01472-f005:**
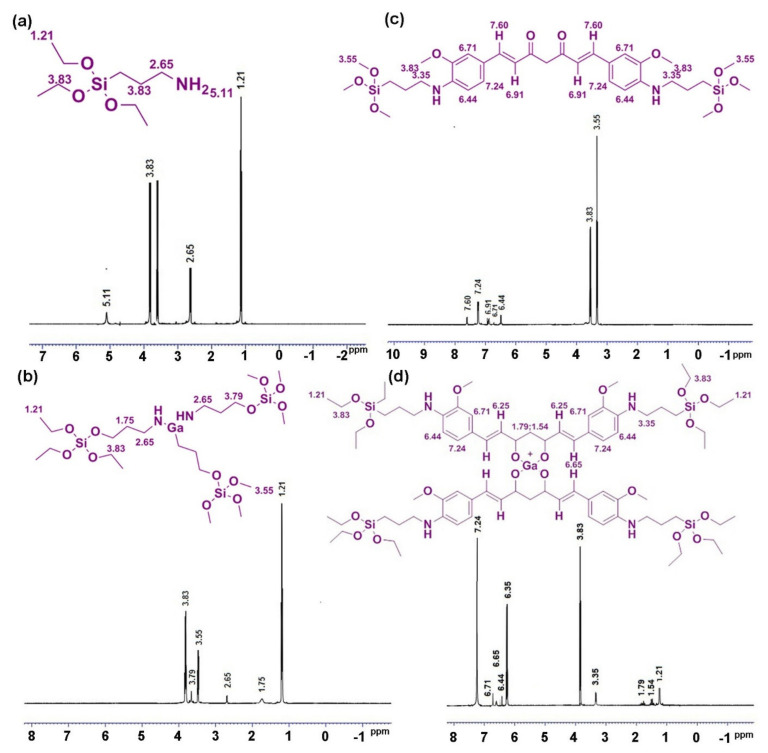
^1^H-NMR spectrum of (**a**) HMSNAP, (**b**) Ga-HMSNAP, (**c**) C-HMSNAP, (**d**) GaC-HMSNAP.

**Figure 6 nanomaterials-12-01472-f006:**
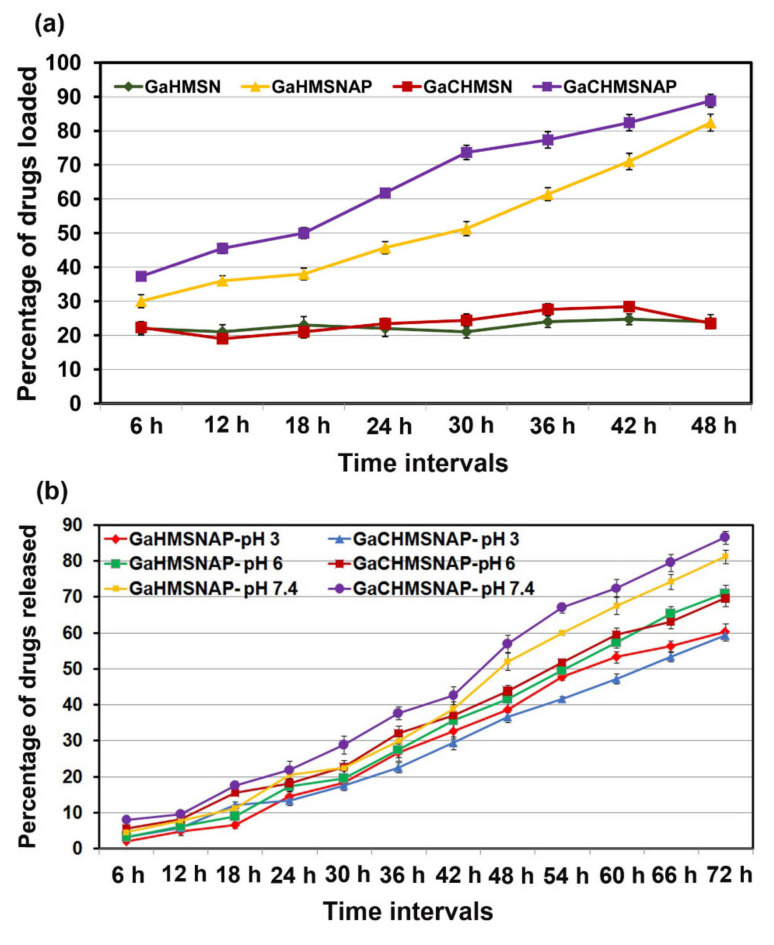
The gallium alone and in a combination of gallium and curcumin loading into HMANP and HMSNAP at different time intervals (**a**) drugs released from HMSNAP at different pH and time intervals (**b**).

**Figure 7 nanomaterials-12-01472-f007:**
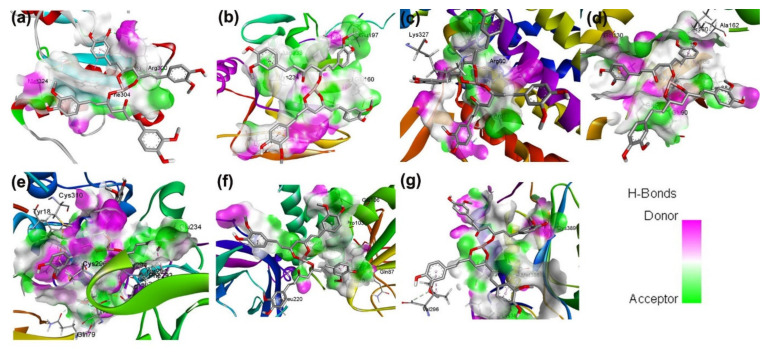
Molecular docking of the gallium-curcumin complex binding site interaction with protein active sites of (**a**) PDI, (**b**) calnexin, (**c**) HSP60, (**d**) PDK, (**e**) Akt1, (**f**) PTEN, and (**g**) caspase. The hydrogen bonding acceptor displayed a green colour and the hydrogen bonding donor a pink colour.

**Figure 8 nanomaterials-12-01472-f008:**
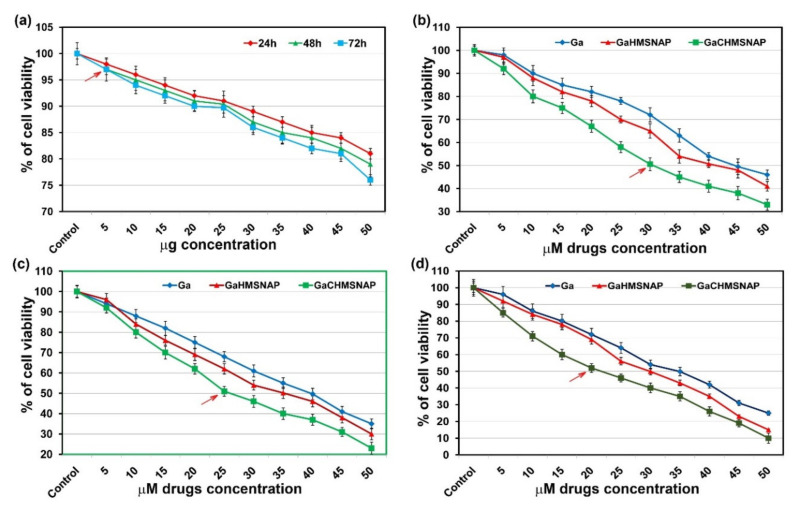
The cell viability assessment of different concentrations of HMSNAP -treated MCF-7 cells. The arrow means all drugs are loaded within the amount of HMSNAP (**a**). Dose and time-dependence of cellular viability due to free gallium, Ga-HMSNAP, and GaC-HMSNAP-treated MCF-7 cells at 24 h (**b**), 48 h (**c**), and 72 h (**d**). Results of MTT are expressed as a percentage related to untreated cells and are set as 100%. The arrow represents IC 50 value of drug-loaded HMSNAP in MCF-7 cells.

**Figure 9 nanomaterials-12-01472-f009:**
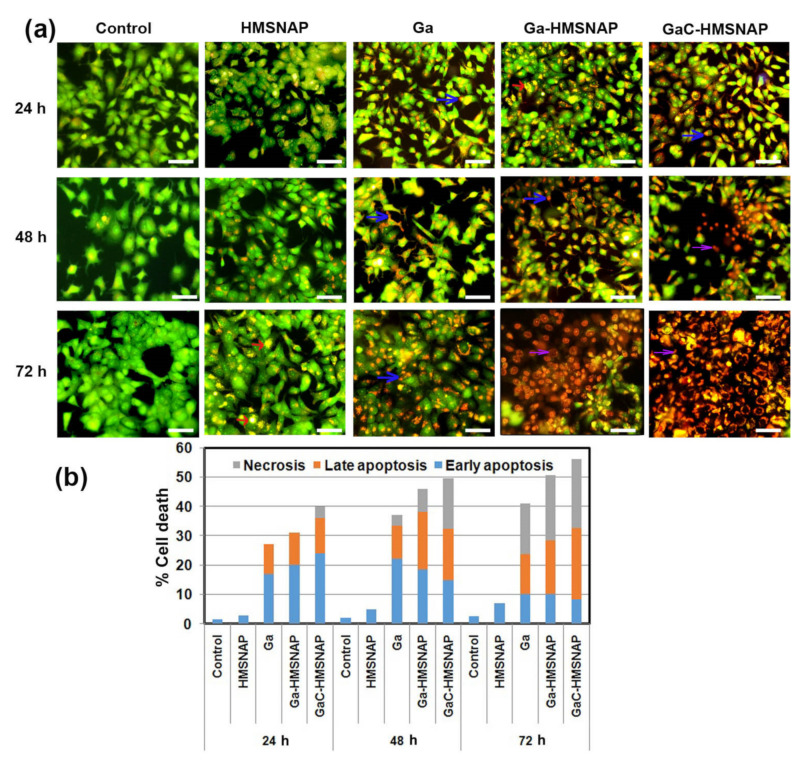
(**a**) Photomicrographs of acridine orange displaying the HMSNAP and drug-loaded HMSNAP-induced cell death in MCF-7 cells at different time intervals. The arrow marks the blue colour, representing early apoptosis, the orange, which represents late apoptosis, and grey, representing necrosis of MCF-7 cells (**b**) The graph represents the percentage of early and late apoptosis and necrosis in HMSNAP and drug-loaded HMSNAP-induced cell death in MCF-7 cells at different time intervals.

**Figure 10 nanomaterials-12-01472-f010:**
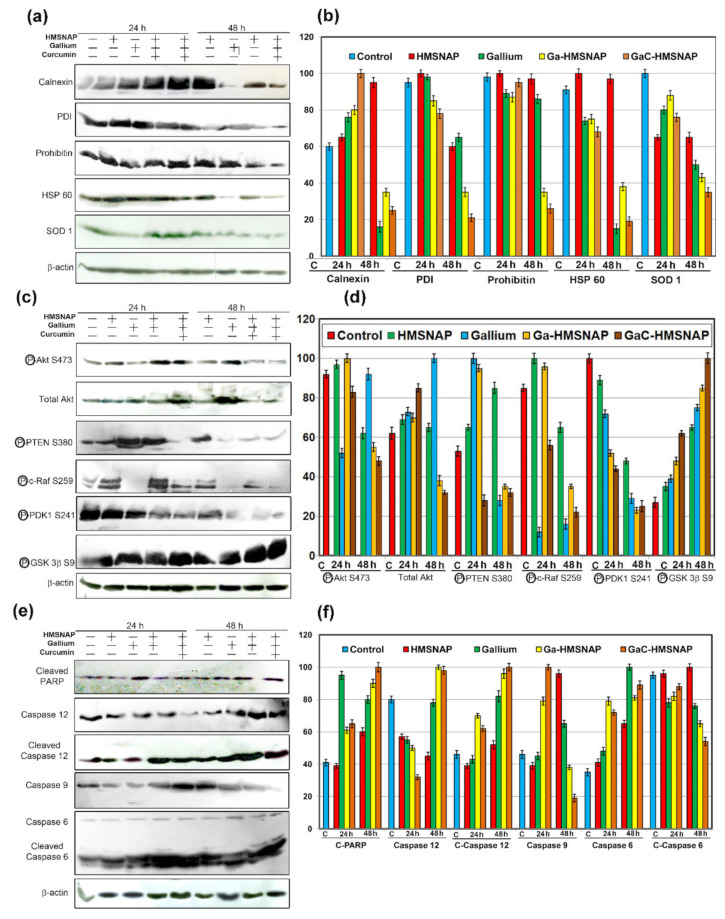
Western blot analysis used to determine the (**a**,**b**) phosphorylation of oncoproteins and tumour-suppressor proteins, (**c**,**d**) ER and mitochondrial proteins, and (**e**,**f**) apoptotic proteins in MCF-7 cells treated with HMSNAP and drug-loaded HMSNAP at 12 h and 24 h intervals, respectively. The graph represents the densitometric analysis of each protein compared with the respective control and calculates the percentage manipulation.

**Table 1 nanomaterials-12-01472-t001:** The percentage of Si, N, O, Ga and C, of HMSNAP and drug-loaded HMSNAP based on XPS results.

Sample	Silica	Nitrogen	Oxygen	Gallium	Carbon
HMSNAP	22.4	0.4	55.8	-	21.4
Ga-HMSNAP	12.8	6	31.6	4.4	45.2
C-HMSNAP	12.4	1.8	39.4	-	46.4
GaC-HMSNAP	13	6.6	32.2	4	44.2

**Table 2 nanomaterials-12-01472-t002:** Textural properties of HMSNAP and drug-loaded HMSNAP.

Sample	A_BET_ (m^2^ g^−1^)	D_BJH_ (nm)	V_BJH_ (cm^3^ g^−1^)
HMSNAP	1067.78	4.45	3.25
Ga-HMSNAP	710.07	3.11	1.49
C-HMSNAP	946.89	3.35	2.48
GaC-HMSNAP	637.39	2.80	1.41

A_BET_, BET surface area; D_BJH_, BJH pore diameter; V_BJH_, BJH volume.

**Table 3 nanomaterials-12-01472-t003:** Drug Release kinetics of HMSNAP and drug-loaded HMSNAP.

Model	Parameter	Gallium Loaded HMSN			Gallium Curcumin Loaded HMSN		
		pH 3	pH 6	pH 7.4	pH 3	pH 6	pH 7.4
Zero order F = K_0_ × t [40]	K_0_	0.014	0.015	0.018	0.013	0.016	0.019
	r^2^	0.9565	0.9561	0.9546	0.9653	0.9697	0.9763
	AIC	66.1659	69.0186	72.7409	61.2602	65.0441	65.7113
First order F = 100 × [1−Exp (−K_1_ × t)] [41]	K_1_	0.000	0.000	0.000	0.000	0.000	0.000
	r^2^	0.8992	0.8763	0.8540	0.9071	0.8860	0.8737
	AIC	76.2562	81.4680	86.7493	73.0921	80.9603	85.7647
Higuchi model F = KH × t1/2 [42]	KH	0.713	0.796	0.927	0.672	0.856	1.027
	r^2^	0.7072	0.6977	0.7004	0.7161	0.7247	0.7436
	AIC	89.0555	92.1877	95.3777	86.4966	91.5440	94.2648
Korsmeyer-Peppas model F = kKP × t^n^ [43]	kKP	0.001	0.001	0.001	0.001	0.001	0.004
	r^2^	0.9854	0.9975	0.9915	0.9965	0.9970	0.9921
	n	1.300	1.382	1.355	1.320	1.296	1.211
	AIC	55.0405	36.7999	54.5763	35.8268	39.4085	54.4488
Hixon-Crowell model F = 100 × [1−(1−_k_HC × t)^3^] [44]	_k_HC	0.000	0.000	0.000	0.000	0.000	0.000
	r^2^	0.9207	0.9044	0.8894	0.9280	0.9159	0.9119
	AIC	73.3839	78.3709	83.4193	70.0364	77.3201	81.4518

AIC = Akaike information criterion, F = fraction of drug release in time t, K_0_ = apparent rate constant of zero order release constant, K_1_ = first order release constant, KH = Higuchi constant, kKP = Korsmeyer–Peppas rate constant, _k_HC = Hixon–Crowell constant, n = diffusional exponent. And r^2^ = Squared correlation coefficient.

## Data Availability

The data presented in this study are available in the article and Appendix A.

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
