# Peer review of "Combinatorial Delivery of Gallium (III) Nitrate and Curcumin Complex-Loaded Hollow Mesoporous Silica Nanoparticles for Breast Cancer Treatment"

_nanomaterials, 2022, doi:10.3390/nano12091472_

Round 1
Reviewer 1 Report
The manuscript submitted by the authors on the topic “ Combinatorial delivery of gallium (III) nitrate and curcumin complex-loaded hollow mesoporous silica nanoparticles for breast cancer treatment” is interesting to the reader and within the scope of the journal.
The following comments are provided below for the author’s attention.
- The introduction should be elaborated more in-depth.
- Discuss in more detail the significance of using mesoporous silica nanoparticles for breast cancer treatment
- On page 1 author has mentions “Hollow mesoporous silica nanoparticles (HMSNP) are considered to be highly versatile and favorable nanoplatforms for drug delivery systems (DDS) because of their assortment, accessibility, and biocompatibility. HMSNP is non-lethal and has enormous explicit surface territory, a tunable pore size, astounding physicochemical steadiness, and artificially modifiable surfaces”. But no single reference is provided to support this sentence.
- Provide an explicit statement of questions being addressed with reference to participants, interventions, comparisons, outcomes, and study design.
- On page 2 section 1. Materials and reagents, kindly provide the grade and purity of all the chemicals used in this work.
- There are several sentences in the script which are really hard to understand. I will suggest the authors should carefully read the script and amend the English language correction throughout the script.
- The synthesis of hollow mesoporous silica nanoparticles (HMSNP) is an acid-catalyzed sol-gel or base-catalyzed sol-gel reaction.
- Kindly provide the stepwise structure mechanism for the synthesis of HMSNAP
- On page 3 author have mentions “The drug release experiments were accomplished with 2 mg mL-1 of 30 μM Ga- HMSNAP, and a combination of 20 μM Gallium and 30 μM curcumin-loaded GaC-HMSNAP was kept separately in phosphate buffer saline (PBS) in three different pH 3, 6, 132 and 7.4.” Did the author have to provide the optimization study for the drug release experiment? If not how the particle value was provided in the text. Need to be clarified.
- Kindly add the following references in the introduction section. [Cancers 13 (14), 3396, 2021; Cancers 13 (9), 2214, 2021;]
- Section 2.4. Drug release kinetics authors have mentioned “The gallium from HMSNAP and the gallium-curcumin complex from HMSNAP release data were separately fitted into five different kinetics models (zero, first, Higuchi, Korsmeyer-Peppas, and Hixson- Crowell) to describe the mechanism of the drug release kinetics and diffusion” I cannot see the single equation of each kinetic models. Kindly provide the expression for all the five kinetics models along with the appropriate references used in Drug release kinetics.
- How did the author performs the zeta potential study. Complete methods and instruments details along with accuracy and preciseness need to be provided.
- After carefully going through the XRD section, I can find that no detailed XRD results were discussed in the script. The author has just mentioned one sentence. “The amorphous nature of HMSNAP was confirmed with XRD (Fig. 2(a)) and expansion of drugs, and the APTES lesion was found in HMSNAP at 10-30 2θ°.” Which is not enough. Need to be more elaborated and explain each graph in detail.
- In FTIR no discussion about the Si-O-Si band?
- In FTIR in the case of HMSNAP and Ga-HMSNAP, Kindly see the C-H band, in reality, such a band is not possible. Kindly check?
- No need for Table 1. The percentage of Si, N, O, Ga and C, of HMSNAP and drug-loaded HMSNAP is based on XPS results. Just remove this table and explain the results in the discussion section.
- How do the surface area, pore volume, and pore diameter impact the drug-loading efficiency and smart delivery vehicle. Need complete discussion.
- Present results of any assessment of the risk of bias across studies
- Provide a general interpretation of the results in the context of other evidence and implications for future research.
Author Response
Reviewer 1
The manuscript submitted by the authors on the topic “Combinatorial delivery of gallium (III) nitrate and curcumin complex-loaded hollow mesoporous silica nanoparticles for breast cancer treatment” is interesting to the reader and within the scope of the journal.
The following comments are provided below for the author’s attention.
We are thankful to the reviewers and editorial board members for the critical evaluation of the manuscript and useful suggestions.
- The introduction should be elaborated more in-depth.
Answer: The introduction part is revised as per the reviewer’s suggestion.
- Discuss in more detail the significance of using mesoporous silica nanoparticles for breast cancer treatment
Answer: As per the reviewer’s suggestion, the details and significance of the mesoporous silica nanoparticles using breast cancer treatment have been included in our revised manuscript.
- On page 1 author has mentions “Hollow mesoporous silica nanoparticles (HMSNP) are considered to be highly versatile and favorable nanoplatform for drug delivery systems (DDS) because of their assortment, accessibility, and biocompatibility. HMSNP is non-lethal and has enormous explicit surface territory, a tunable pore size, astounding physicochemical steadiness, and artificially modifiable surfaces”. But no single reference is provided to support this sentence.
Answer: We appreciate the reviewer for his/her constructive comment. We have incorporated relevant references in our revised manuscript. For your kind information, the below references are included.
- Harini L, Bose K, Viswanathan TM, Kumar NS, Sundar K, Kathiresan T (2021) Mesoporous Silica Nanoparticles Are Nanocarrier for Drug Loading and Induces Cell Death in Breast Cancer. In: Environmental Biotechnology Volume 4. Springer, pp 225-245
- Huang L, Liu J, Gao F, Cheng Q, Lu B, Zheng H, Xu H, Xu P, Zhang X, Zeng X (2018) A dual-responsive, hyaluronic acid targeted drug delivery system based on hollow mesoporous silica nanoparticles for cancer therapy. Journal of Materials Chemistry B 6 (28):4618-4629
- Provide an explicit statement of questions being addressed with reference to participants, interventions, comparisons, outcomes, and study design.
Answer: We thank the reviewers for his/her critical reviews of our manuscript. Before designing this research, we have done a comprehensive review and literature search was conducted in PubMed, Scopus, Embase, Web of Science, Google Scholar and National library of medicine (NLM). We have identified more than 50 references. Reference lists were scrutinized, and citation searches were performed on the included studies. The primary outcome was the quality of literature searches and the secondary outcome was time spent on the literature search when the PICO model was used as a search strategy tool, compared to the use of another conceptualizing tool or unguided searching.
- On page 2 section Materials and reagents, kindly provide the grade and purity of all the chemicals used in this work.
Answer: As per the reviewer’s suggestion, the grade and purity of the used materials and reagents were incorporated in our revised manuscript.
- There are several sentences in the script which are really hard to understand. I will suggest the authors should carefully read the script and amend the English language correction throughout the script.
Answers: We are sorry for the inadvertent grammar mistakes. The English language of the revised manuscript has been revised and edited by a language expert.
- The synthesis of hollow mesoporous silica nanoparticles (HMSNP) is an acid-catalyzed sol-gel or base-catalyzed sol-gel reaction.
Answer: For your kind information, the hollow mesoporous silica nanoparticles (HMSNP) was synthesized through a base-catalyzed sol-gel reaction, because we are using ammonia (moderately basic) as a catalyst.
- Kindly provide the stepwise structure mechanism for the synthesis of HMSNAP
Answer: As per the reviewer’s suggestion, we have included stepwise structure mechanisms for the synthesis of HMSNAP in our revised manuscript.
- On page 3 author have mentions “The drug release experiments were accomplished with 2 mg mL-1 of 30 μM Ga- HMSNAP, and a combination of 20 μM Gallium and 30 μM curcumin-loaded GaC-HMSNAP was kept separately in phosphate buffer saline (PBS) in three different pH 3, 6, and 7.4.” Did the author have to provide the optimization study for the drug release experiment? If not how the particle value was provided in the text. Need to be clarified.
Answer: As per our previously published articles, we have chosen the concentration of HMSNAP for drug release studies. Actually, we don’t know the exact pH (buffer) for the highest release of loaded drugs from HMSNAP. Therefore, we selected three different pH (3, 6, and 7.4) for our HMSNAP drug delivery experiments. The synthesized HMSNAP were uniform in size, so they did not change/influence the release studies.
- Kindly add the following references in the introduction section. [Cancers 13 (14), 3396, 2021; Cancers 13 (9), 2214, 2021]
Answer: As per the reviewer’s suggestion, the above-mentioned references has been included in our revised manuscript.
- Section 2.4. Drug release kinetics authors have mentioned “The gallium from HMSNAP and the gallium-curcumin complex from HMSNAP release data were separately fitted into five different kinetics models (zero, first, Higuchi, Korsmeyer-Peppas, and Hixson- Crowell) to describe the mechanism of the drug release kinetics and diffusion” I cannot see the single equation of each kinetic models. Kindly provide the expression for all the five kinetics models along with the appropriate references used in Drug release kinetics.
Answer: Thank you for your critical viewpoints. We have already mentioned all the model equations in the drug release kinetics in Table 3.
- How the author did performs the zeta potential study. Complete methods and instruments details along with accuracy and preciseness need to be provided.
Answer: The HMSNAP, Ga-HMSNAP, and GaC-HMSNAP are separately dispersed in water and analysed for their size and surface charge through DLS and zeta potential by using the Nanoparticle analyser SZ-100 (Horiba, Kyoto, Japan). The particle size and zeta potential in liquid suspension were measured at 25 °C using Nanoparticle analyser SZ-100 (Horiba, Kyoto, Japan). A 633 nm, He-Ne laser was used as the light source, while an avalanche photodiode (APD) served as the detector. Particle size was measured using dynamic light scattering method, where the scattered light was collected at 173°.
- After carefully going through the XRD section, I can find that no detailed XRD results were discussed in the script. The author has just mentioned one sentence. “The amorphous nature of HMSNAP was confirmed with XRD (Fig. 2(a)) and expansion of drugs, and the APTES lesion was found in HMSNAP at 10-30 2θ°.” Which is not enough. Need to be more elaborated and explain each graph in detail.
Answer: Thank you for your valuable suggestions to improve our manuscript. We have incorporated more details about the XRD results in our revised manuscript.
- In FTIR no discussion about the Si-O-Si band?
Answer: Thank you for your valuable suggestions to improve our manuscript. In the HMSNAP the most intensive band Si-O-Si bending present in the range of 500 - 800 cm-1. The peak value correspond to the symmetric and anti-symmetric stretching vibration of the Si–O–Si. The above-mentioned comments has been included in the revised manuscript.
- In FTIR in the case of HMSNAP and Ga-HMSNAP, Kindly see the C-H band, in reality, such a band is not possible. Kindly check?
Answer: We agree with the comment. The C-H band was one of the band to confirm the amine functionalization of HMSNAP. In the hollow mesoporous silica nanoparticles was synthesis, when the HMSNAP was functionalized with APTES, in addition, the peak values in the range of 1000 to 1250 cm-1 and near 2900 cm-1 are associated with mesoporous silica.
- No need for Table 1. The percentage of Si, N, O, Ga and C, of HMSNAP and drug-loaded HMSNAP is based on XPS results. Just remove this table and explain the results in the discussion section.
Answer: Sorry sir, the Table 1 is very important for this manuscript. Because, we provide details about the% elemental report. Moreover, observed results also explained in results and discussion section.
- How do the surface area, pore volume, and pore diameter impact the drug-loading efficiency and smart delivery vehicle? Need complete discussion.
Answer: Thank you for your valuable comments. The mentioned details and their discussion was incorporated in our revised manuscript.
- Present results of any assessment of the risk of bias across studies
Answer: We agree with the comment. There is no risk of bias across studies of the present result.
- Provide a general interpretation of the results in the context of other evidence and implications for future research.
Answer: Thank you for your valuable suggestions to improve our manuscript. Now included interpretations of all results obtained from our present and future research works.

Reviewer 2 Report
This manuscript studied the co-delivery efficacy of gallium (III) nitrate and curcumin using hollow mesoporous silica nanoparticles for breast cancer treatment. By analyzing their interactions and the mechanism of gallium-curcumin complex-induced cancer cell death, it was found that GaC-HMNSAP induced cell death through the mitochondria-intrinsic cell death pathway. In my opinion, this manuscript can be considered for publication in Nanomaterials after addressing the modifications.
- The description of mesoporous silica in the Abstract part was too much, which is not consistent with the main points of the whole text. Please adjust appropriately.
- The drug was released mostly at pH 7.4 when compared to the parallel group. So how to address the safety of the nanomaterials. And, the reason for choosing pH 3 and 6 should be provided.
- Please add the standard card of the corresponding element to the XRD diagram.
- Please standardize the basic format of the graphs, such as borders, axes, etc.
- Was there any other data to support the synergistic effect of gallium and curcumin treatment?
- The experiments in vitro were not enough. The cell uptake and apoptosis analysis data are suggested to be added.
Author Response
Reviewer 2
Comments and Suggestions for Authors
This manuscript studied the co-delivery efficacy of gallium (III) nitrate and curcumin using hollow mesoporous silica nanoparticles for breast cancer treatment. By analyzing their interactions and the mechanism of gallium-curcumin complex-induced cancer cell death, it was found that GaC-HMNSAP induced cell death through the mitochondria-intrinsic cell death pathway. In my opinion, this manuscript can be considered for publication in Nanomaterials after addressing the modifications.
- The description of mesoporous silica in the Abstract part was too much, which is not consistent with the main points of the whole text. Please adjust appropriately.
Answer: Thank you for your valuable comments. As per the suggestion received, we re-written the abstract and the same was incorporated in our revised manuscript.
- The drug was released mostly at pH 7.4 when compared to the parallel group. So how to address the safety of the nanomaterials. And, the reason for choosing pH 3 and 6 should be provided.
Answer: In the drug release study, we want to confirm the exact release pH of the nanomaterials. So, we focused on three different pH like 3, 6 and 7.4 and the highest percentage of drug molecules released in the pH of 7.4. PBS at pH 6 and 7.4 are commonly used to mimic the colon, PBS buffer pH 3 to mimic the stomach environment. Therefore we choose three different pH buffers used in our study.
- Please add the standard card of the corresponding element to the XRD diagram.
Answer: Thank you for your valuable comments. The standard card has been added in the XRD diagram of revised manuscript. JCDS pattern of amorphous mesoporous silica nanoparticles was 41-1144, this was also included in our revised manuscript.
- Please standardize the basic format of the graphs, such as borders, axes, etc.
Answer: Thank you for your critical review. All the graphs are corrected in our revised manuscript.
- Was there any other data to support the synergistic effect of gallium and curcumin treatment?
Answer: We agree with the comment. All relevant supporting data were included in the revised manuscript.
- The experiments in vitro were not enough. The cell uptake and apoptosis analysis data are suggested to be added.
Answer: We agree with the comment. The apoptosis analysis of HMSNAP, Ga-HMSNAP, and GaC-HMSNAP has been presented and discussed in our revised manuscript.

Round 2
Reviewer 1 Report
I cannot see any change made by the authors in the revised manuscript. Highlight the correction made in the revised manuscript. If the author cannot able to incorporate any change they should clearly mention the reason for it. Resubmit the revised highlighted manuscript for review evaluation. Authors should carefully make all the changes. Each and every point will be thoroughly reviewed by the reviewer.
Author Response
The manuscript submitted by the authors on the topic “Combinatorial delivery of gallium (III) nitrate and curcumin complex-loaded hollow mesoporous silica nanoparticles for breast cancer treatment” is interesting to the reader and within the scope of the journal.
The following comments are provided below for the author’s attention.
We are thankful to the reviewers and editorial board members for the critical evaluation of the manuscript and useful suggestions.
- The introduction should be elaborated more in-depth.
Answer: The introduction part is revised as per the reviewer’s suggestion (Page No. 3; line no: 58-78).
- Discuss in more detail the significance of using mesoporous silica nanoparticles for breast cancer treatment
Answer: As per the reviewer’s suggestion, the details and significance of the mesoporous silica nanoparticles using breast cancer treatment have been included in our revised manuscript. (Page No. 21; line no: 474-478).
- The enormous potential of hollow mesoporous silica nanoparticles (HMSN) has recently achieved more interest in multiple drug loading capacity due to their effective attraction in nanomedicine platform especially in breast cancer treatment. [1]
- The hollow mesoporous silica nanoparticles induce the in vitro cytotoxicity in tumor cells and their effective killing mechanisms is one of the significance to confirm the potential of hollow mesoporous silica nanoparticles in breast cancer therapy. [2]
- The significance in traditional cancer medicine with hollow mesoporous silica nanoparticles is highly conventional and non-toxicity. In the breast cancer, the hollow mesoporous silica nanoparticles used to overcome the formal issues which can applied in effective attention of significant in hollow mesoporous silica nanoparticles. [3]
- Li, Y., Li, N., Pan, W., Yu, Z., Yang, L., & Tang, B. (2017). Hollow mesoporous silica nanoparticles with tunable structures for controlled drug delivery. ACS applied materials & interfaces, 9(3), 2123-2129.
- Gautam, M., Thapa, R. K., Poudel, B. K., Gupta, B., Ruttala, H. B., Nguyen, H. T., ... & Kim, J. O. (2019). Aerosol technique-based carbon-encapsulated hollow mesoporous silica nanoparticles for synergistic chemo-photothermal therapy. Acta Biomaterialia, 88, 448-461.
- Niu, S., Zhang, X., Williams, G. R., Wu, J., Gao, F., Fu, Z., ... & Zhu, L. M. (2021). Hollow mesoporous silica nanoparticles gated by chitosan-copper sulfide composites as theranostic agents for the treatment of breast cancer. Acta Biomaterialia, 126, 408-420.
- On page 1 author has mentions “Hollow mesoporous silica nanoparticles (HMSNP) are considered to be highly versatile and favorable nanoplatform for drug delivery systems (DDS) because of their assortment, accessibility, and biocompatibility. HMSNP is non-lethal and has enormous explicit surface territory, a tunable pore size, astounding physicochemical steadiness, and artificially modifiable surfaces”. But no single reference is provided to support this sentence.
Answer: We appreciate the reviewer for his/her constructive comment. We have incorporated relevant references in our revised manuscript. For your kind information, the below references are included. (Page No. 3; line no: 58-78)
- Harini L, Bose K, Viswanathan TM, Kumar NS, Sundar K, Kathiresan T (2021) Mesoporous Silica Nanoparticles Are Nanocarrier for Drug Loading and Induces Cell Death in Breast Cancer. In: Environmental Biotechnology Volume 4. Springer, pp 225-245
- Huang L, Liu J, Gao F, Cheng Q, Lu B, Zheng H, Xu H, Xu P, Zhang X, Zeng X (2018) A dual-responsive, hyaluronic acid targeted drug delivery system based on hollow mesoporous silica nanoparticles for cancer therapy. Journal of Materials Chemistry B 6 (28):4618-4629
- Provide an explicit statement of questions being addressed with reference to participants, interventions, comparisons, outcomes, and study design.
Answer: We thank the reviewers for his/her critical reviews of our manuscript. Before designing this research, we have done a comprehensive review and literature search was conducted in PubMed, Scopus, Embase, Web of Science, Google Scholar and National library of medicine (NLM). We have identified more than 50 references. Reference lists were scrutinized, and citation searches were performed on the included studies. The primary outcome was the quality of literature searches and the secondary outcome was time spent on the literature search when the PICO model was used as a search strategy tool, compared to the use of another conceptualizing tool or unguided searching.
- On page 2 section Materials and reagents, kindly provide the grade and purity of all the chemicals used in this work.
Answer: As per the reviewer’s suggestion, the grade and purity of the used materials and reagents were incorporated in our revised manuscript. (Page No. 5,6; line no: 120-133)
- There are several sentences in the script which are really hard to understand. I will suggest the authors should carefully read the script and amend the English language correction throughout the script.
Answers: We are sorry for the inadvertent grammar mistakes. The English language of the revised manuscript has been revised and edited by a language expert.
- The synthesis of hollow mesoporous silica nanoparticles (HMSNP) is an acid-catalyzed sol-gel or base-catalyzed sol-gel reaction.
Answer: For your kind information, the hollow mesoporous silica nanoparticles (HMSNP) was synthesized through a base-catalyzed sol-gel reaction, because we are using ammonia (moderately basic) as a catalyst.
- Kindly provide the stepwise structure mechanism for the synthesis of HMSNAP
Answer: As per reviewer’s suggestion, we include stepwise structure mechanism the synthesis of HMSNAP in revised manuscript.
Deposition of Silica Layer on Surfactant and Subsequent removal of the surfactant by calcination has been known to be a general approach for preparation of HMSNP.
- The surfactant was mixed with aqueous solution which contain water and ethanol.
- Then, TEOS addition develop the condensation over the micelles and ammonia induce catalytic process to form silica.
- Subsequently, remove the surfactant by calcination to produce hollow structure in the nanoparticles to form HMSNP.
- Finally, the decoration of nanoparticles with APTES on the surface of nanoparticles to form HMSNAP.
- On page 3 author have mentions “The drug release experiments were accomplished with 2 mg mL-1 of 30 μM Ga- HMSNAP, and a combination of 20 μM Gallium and 30 μM curcumin-loaded GaC-HMSNAP was kept separately in phosphate buffer saline (PBS) in three different pH 3, 6, and 7.4.” Did the author have to provide the optimization study for the drug release experiment? If not how the particle value was provided in the text. Need to be clarified.
Answer: As per our previously published articles, we have chosen the concentration of HMSNAP for drug release studies. Actually, we don’t know the exact pH (buffer) for the highest release of loaded drugs from HMSNAP. Therefore, we selected three different pH (3, 6, and 7.4) for our HMSNAP drug delivery experiments. The synthesized HMSNAP were uniform in size, so they did not change/influence the release studies.
- Kindly add the following references in the introduction section. [Cancers 13 (14), 3396, 2021; Cancers 13 (9), 2214, 2021] (Page No. 3; line no: 7,8)
Answer: As per the reviewer’s suggestion, the above-mentioned references has been included in our revised manuscript.
- Section 2.4. Drug release kinetics authors have mentioned “The gallium from HMSNAP and the gallium-curcumin complex from HMSNAP release data were separately fitted into five different kinetics models (zero, first, Higuchi, Korsmeyer-Peppas, and Hixson- Crowell) to describe the mechanism of the drug release kinetics and diffusion” I cannot see the single equation of each kinetic models. Kindly provide the expression for all the five kinetics models along with the appropriate references used in Drug release kinetics.
Answer: Thank you for your critical viewpoints. We have already mentioned all the model equations in the drug release kinetics in Table 3.
- How the author did performs the zeta potential study. Complete methods and instruments details along with accuracy and preciseness need to be provided.
Answer: The HMSNAP, Ga-HMSNAP, and GaC-HMSNAP are separately dispersed in water and analysed for their size and surface charge through DLS and zeta potential by using the Nanoparticle analyser SZ-100 (Horiba, Kyoto, Japan). The particle size and zeta potential in liquid suspension were measured at 25 °C using Nanoparticle analyser SZ-100 (Horiba, Kyoto, Japan). A 633 nm, He-Ne laser was used as the light source, while an avalanche photodiode (APD) served as the detector. Particle size was measured using dynamic light scattering method, where the scattered light was collected at 173°.
- After carefully going through the XRD section, I can find that no detailed XRD results were discussed in the script. The author has just mentioned one sentence. “The amorphous nature of HMSNAP was confirmed with XRD (Fig. 2(a)) and expansion of drugs, and the APTES lesion was found in HMSNAP at 10-30 2θ°.” Which is not enough. Need to be more elaborated and explain each graph in detail.
Answer: Thank you for your valuable suggestions to improve our manuscript. We have incorporated more details about the XRD results in our revised manuscript.
- In FTIR no discussion about the Si-O-Si band?
Answer: Thank you for your valuable suggestions to improve our manuscript. In the HMSNAP the most intensive band Si-O-Si bending present in the range of 500 - 800 cm-1. The peak value correspond to the symmetric and anti-symmetric stretching vibration of the Si–O–Si. The above-mentioned comments has been included in the revised manuscript.
- In FTIR in the case of HMSNAP and Ga-HMSNAP, Kindly see the C-H band, in reality, such a band is not possible. Kindly check?
Answer: We agree with the comment. The C-H band was one of the band to confirm the amine functionalization of HMSNAP. In the hollow mesoporous silica nanoparticles was synthesis, when the HMSNAP was functionalized with APTES, in addition, the peak values in the range of 1000 to 1250 cm-1 and near 2900 cm-1 are associated with mesoporous silica.
- No need for Table 1. The percentage of Si, N, O, Ga and C, of HMSNAP and drug-loaded HMSNAP is based on XPS results. Just remove this table and explain the results in the discussion section.
Answer: Sorry sir, the Table 1 is very important for this manuscript. Because, we provide details about the% elemental report. Moreover, observed results also explained in results and discussion section.
- How do the surface area, pore volume, and pore diameter impact the drug-loading efficiency and smart delivery vehicle? Need complete discussion.
Answer: Thank you for your valuable comments. The mentioned details and their discussion was incorporated in our revised manuscript.
- Present results of any assessment of the risk of bias across studies
Answer: We agree with the comment. There is no risk of bias across studies of the present result.
- Provide a general interpretation of the results in the context of other evidence and implications for future research.
Answer: Thank you for your valuable suggestions to improve our manuscript. Now included interpretations of all results obtained from our present and future research works.

Reviewer 2 Report
This manuscript can be considered for publication in its current form.
Author Response
Answer: Many thanks for your critical review and valuable suggestions.

Round 3
Reviewer 1 Report
The authors have revised the manuscript.